# BSTT: A Bayesian Spatial-Temporal Transformer for Sleep Staging

**Yuchen Liu**
Institute of Automation, CAS
University of Chinese Academy of Sciences
Institute of Computing Technology, CAS
`yuchen.liu.eric@outlook.com`

**Ziyu Jia**[*]
Institute of Automation, CAS
University of Chinese Academy of Sciences
`jia.ziyu@outlook.com`

## ABSTRACT

Sleep staging is helpful in assessing sleep quality and diagnosing sleep disorders. However, how to adequately capture the temporal and spatial relations of the brain during sleep remains a challenge. In particular, existing methods cannot adaptively infer spatial-temporal relations of the brain under different sleep stages. In this paper, we propose a novel Bayesian spatial-temporal relation inference neural network, named Bayesian spatial-temporal transformer (BSTT), for sleep staging. Our model is able to adaptively infer brain spatial-temporal relations during sleep for spatial-temporal feature modeling through a well-designed Bayesian relation inference component. Meanwhile, our model also includes a spatial transformer for extracting brain spatial features and a temporal transformer for capturing temporal features. Experiments show that our BSTT outperforms state-of-the-art baselines on ISRUC and MASS datasets. In addition, the visual analysis shows that the spatial-temporal relations obtained by BSTT inference have certain interpretability for sleep staging.

## 1 INTRODUCTION

Sleep staging is essential for assessing sleep quality and diagnosing sleep disorders. Sleep specialists typically classify sleep stages based on the AASM sleep standard and polysomnography (PSG) recordings to aid in diagnosis. The AASM standard not only provides criteria for determining each sleep period, but also documents conversion rules between different sleep stages, which is known as sleep transition rules, to help sleep specialists identify sleep stages when sleep transitions occur. However, artificial sleep staging takes a long time, and the classification results are greatly affected by professional level and subjectivity (Supratak et al., 2017). Therefore, automatic classification methods are applied into sleep staging to improve efficiency.

Traditional machine learning methods use artificially designed features for sleep staging, which improves the efficiency of staging to a certain extent (Fraiwan et al., 2012). However, the accuracy of traditional machine learning methods relies heavily on feature engineering and feature selection, which still requires a lot of expert knowledge. To address the above problems, deep learning methods have been applied to sleep staging and achieved satisfactory classification performance (Phan et al., 2019; Jia et al., 2022a;b). Most of the early deep learning methods focus on the temporal information of the sleep data, utilizing convolutional neural networks (CNN) and recurrent neural networks (RNN) to capture temporal features for sleep staging (Jain & Ganesan, 2021; Perslev et al., 2019). In addition, some studies have shown that the spatial topology of the brain behave differently in different sleep stages (Khanal, 2019), which means that both the temporal and spatial relations of the brain are both important during sleep. Therefore, some researchers try to use the spatial and temporal characteristics of the brain for sleep staging (Jia et al., 2020b; Phan et al., 2022; Jia et al., 2020a). Although the above methods achieve good classification performance, it is challenging to model spatial and temporal relations. Specifically, for the modeling of temporal relations, some approaches attempt to capture sleep transition rules in sleep to serve the identification of specific sleep stages. However, it is difficult for these methods to explicitly demonstrate the relation of

---

[*]Corresponding author.

different sleep time slices in accordance with the AASM sleep standard. Besides, for the modeling of spatial relations, spatial convolution operation is employed to extract the spatial features of the brain, which is insufficient that it may ignore the spatial topology of the brain by most methods (Zhou et al., 2021a; Perslev et al., 2019). A few researches utilize spatial topology and temporal relation information of brain for sleep staging by graph convolutional networks, but the constructed brain networks still lack interpretability to a certain extent (Jia et al., 2020b).

To address the above challenges, we propose a novel model called Bayesian spatial-temporal transformer (BSTT) for sleep staging. The proposed model integrates the transformer and Bayesian relation inference in a unified framework. Specifically, we design the spatial-temporal transformer architecture, which can capture the temporal and spatial features of the brain. Besides, we propose the Bayesian relational inference component which comes in two forms, Bayesian temporal relation inference and Bayesian spatial relation inference. Wherefore, it can infer the spatial-temporal relations of objects and generate the relation intensity graphs. Specifically, the main contributions of our BSTT are summarized as follows:

- We design Bayesian relational inference component which can adaptively infer spatial-temporal relations of brain during sleep in the service of capturing spatial-temporal relations.

- We apply the spatial-temporal transformer architecture to simultaneously model spatial-temporal relations. It can effectively capture the spatial-temporal features of the brain and enhance the model's ability to model spatial-temporal relations.

- Experimental results show that the proposed BSTT achieves the state-of-the-art in multiple sleep staging datasets. The visual analysis shows that our model has a certain degree of interpretability for sleep staging.

## 2 RELATED WORK

Identifying sleep stages plays an important role in diagnosing and treating sleep disorders. Earlier, the support vector machine (SVM) and random forest (RF) are used for sleep staging (Fraiwan et al., 2012). However, these methods need hand-crafted features, which require a lot of prior knowledge. Currently, deep learning methods have become the primary method for sleep staging.

Early deep learning methods extract temporal features of sleep signals for classification. The earliest methods are based on the CNN models (Tsinalis et al., 2016; Chambon et al., 2018). For example, Chambon et al. propose a convolutional neural network that can extract temporal-invariant features from sleep signals (Chambon et al., 2018). Furthermore, Eldele et al. develop a multi-resolution CNN with adaptive feature recalibration to extract representative features (Eldele et al., 2021). In addition, RNN models have been gradually used for sleep staging (Phan et al., 2019; Perslev et al., 2019; Phan et al., 2018). For example, Phan et al. propose a deep bidirectional RNN model with attention mechanism for single-channel EEG (Phan et al., 2018). They then design an end-to-end hierarchical RNN architecture for capturing different levels of EEG signal features (Phan et al., 2019). Some studies combine CNN with RNN (Supratak & Guo, 2020; Guillot & Thorey, 2021; Dong et al., 2017). For example, Suratak et al. propose a hybrid model combining CNN and RNN to extract rich temporal features (Supratak et al., 2017). In addition, Phan et al. introduce transformer into the sleep staging task to capture the temporal context features of sleep signals (Phan et al., 2022). Jia et al. design a fully convolutional model to capture the typical waveform of sleep signals (Jia et al., 2021b).

Further, several studies have shown the importance of brain spatial relations for sleep staging (Khanal, 2019; Sakkalis, 2011). Some researchers try to model the spatial-temporal characteristics of sleep data. For example, Jia et al. propose an adaptive deep learning model for sleep staging. The proposed spatial-temporal graph convolutional network is used to extract spatial features and capture transition rules (Jia et al., 2020b). They also propose a multi-view spatial-temporal graph convolutional network based on domain generalization, which models the multi-view-based spatial characteristics of the brain (Jia et al., 2021a).

Although the above models achieve good classification performance, these models do not adequately model spatial-temporal properties or effectively reason and capture spatial-temporal rela-

tions. Therefore, our method attempts to model spatial-temporal relations using Bayesian inference, combined with state-of-the-art transformer architectures for sleep staging.

## 3 PRELIMINARIES

The proposed model processes data from successive $T$ sleep epochs and predicts the label of the epoch in the middle. Each sleep epoch is defined as $x \in \mathbb{R}^{C \times N}$, where $C$ represents the number of channels of the sleep epoch (i.e. the EEG channel in this work. Since EEG signals from different channels are extracted from different regions of the brain, spatial relations are contained among these channels.) and $N$ represents the number of sampling points in a sleep epoch. The input sequence of sleep epochs is defined as $\boldsymbol{x} = \{x_1, x_2, \ldots, x_T\}$, where $x_i$ denotes a sleep epoch ($i \in [1, 2, \ldots, T]$) and $T$ is the number of sleep epochs.

The sleep staging problem is defined as: learning an artificial neural network $\boldsymbol{F}$ based on Bayesian spatial-temporal transformer, which can infer the spatial-temporal relations of the input sleep epoch sequence $\boldsymbol{x}$ and map it to the corresponding sleep stage $\widehat{Y}$, where $\widehat{Y}$ is the classification result of the middle epoch $x_{\mathrm{m}}$. According to the AASM standard, each $\widehat{Y} \in \{0, 1, 2, 3, 4\}$ is matched to five sleep stages W, N1, N2, N3, and REM, respectively.

## 4 BAYESIAN SPATIAL-TEMPORAL TRANSFORMER

We propose a novel model named Bayesian spatial-temporal transformer (BSTT) for sleep staging. The core ideas of our model are summarized as follows:

- Infer the spatial-temporal relations of the brain based on Bayesian inference method.
- Design the Bayesian transformer architecture while capturing the spatial-temporal features of the brain.
- Integrate Bayesian relational inference components and transformer architecture into a unified framework which can stage sleep period effectively.

The overall model is carefully designed to accurately classify different sleep stages.

### 4.1 ARCHITECTURE

The overall architecture of the proposed BSTT is shown in Figure 1. The EEG signals are first encoded by the embedding layer. The spatial-temporal relations are then inferred and modeled by Bayesian spatial-temporal transformer module. Specifically, the Bayesian spatial-temporal transformer includes a Bayesian spatial transformer and a Bayesian temporal transformer. The Bayesian spatial transformer can reason about spatial relations in the brain and capture spatial features. The Bayesian temporal transformer can reason about the temporal relations of consecutive sleep epochs and capture temporal features. Finally, the predictions of different sleep stages are performed by the classification layer.

#### 4.1.1 BAYESIAN RELATION INFERENCE

Capturing the spatial-temporal relations of brain signals during sleep can better serve sleep staging task. However, due to the difficulty in inferring the spatial-temporal relations of sleep, current researches are insufficient for modeling the spatial-temporal relations. Inspired by deep graph random process (DGP) proposed in recent research (Huang et al., 2020), we propose the Bayesian relation inference method. Bayesian relational inference is the core component of our model, which can infer relations between each pair of object nodes and build relation intensity graphs. In this article, an object node represents the embedding of an EEG channel or the embedding of a certain time slice. The construction of the relation intensity graph is mainly divided into the following steps:

*Step 1: Edge embedding initialization.* The input of the Bayesian relational inference is the node embeddings of the object. Building a sleep relation intensity graph starts with generating edge embeddings. Specifically, the node embeddings of the object are spliced and generated into edge

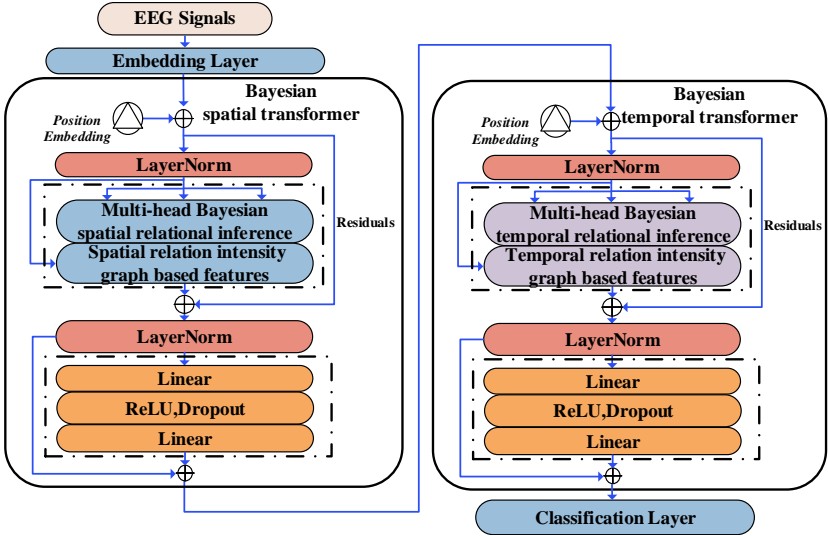

Figure 1: The overall architecture of the proposed Bayesian spatial-temporal transformer for sleep staging. BSTT includes two Bayesian transformer modules, a spatial Bayesian transformer and a temporal Bayesian transformer. For each transformer module, the input features are passed through the position embedding and layernorm layer. Then the multi-head Bayesian relation inference component infers the object's spatial or temporal relation and captures the spatial-temporal features. The residual connection is used to prevent overfitting and gradient disappearance. Here, $\oplus$ means add.

embeddings. We first apply a linear neural network $\boldsymbol{f}_\theta$ to generate edge embeddings:

$$\boldsymbol{E}_{\mathrm{e}} = \boldsymbol{f}_\theta\left(\boldsymbol{E}_{\mathrm{n}}\left[i,j\right]\right) \quad (i,j \in [1,n] \ \&\& \ i \mathrel{!}= j) \tag{1}$$

where $\boldsymbol{E}_{\mathrm{n}} \in \mathbb{R}^{B \times N_v \times V}$ represents the input $n$ node objects, $\boldsymbol{E}_{\mathrm{n}}[i,j] \in \mathbb{R}^{B \times N_e \times 2V}$ represents the splicing of two node embeddings, and $\boldsymbol{E}_{\mathrm{e}} \in \mathbb{R}^{B \times N_e \times E}$ represents the generated edge embeddings.

*Step 2: Edge embedding coupling.* Coupling is one of the key steps in relation inference, the purpose is to obtain a summary graph of the relation between object nodes (Huang et al., 2020). We assume that the edges of the summary graph $\ddot{M}_{i,j}$ are a summary of an $n \to \infty$ of $\lambda \to 0$ Binomial distributions, which means $\ddot{m}_{i,j} \sim \mathcal{B}(n,\lambda)$, due to the uncertainty of spatial-temporal relations of brain. Drawing from the Virtual Recurrent Neural Network (VRNN) model, the parameters of the approximate posteriors are estimated using a Recurrent Neural Network (RNN) to encode features. However, in contrast to VRNN, Bayesian relation inference component contains an approximate posterior $q(\ddot{m}|E_e)$, whose inference and sampling cannot be solved in a computationally feasible manner due to its infinite $n$. By De MoivreLaplace theorem (Sheynin, 1977) and DGP (Huang et al., 2020), we can subject these edge embeddings to a coupling transformation as follows:

$$n_{\mathrm{i,j}} = \zeta\left(\mathrm{L_{mean}}\left(\boldsymbol{E}_{\mathrm{e_{i,j}}}\right)\right) + \epsilon \tag{2}$$

$$\widetilde{\sigma}_{\mathrm{i,j}} = \zeta\left(\mathrm{L_{std}}\left(\boldsymbol{E}_{\mathrm{e_{i,j}}}\right)\right) \tag{3}$$

$$m_{\mathrm{i,j}} = \frac{1 + 2n_{\mathrm{i,j}}\,\tilde{\sigma}_{\mathrm{i,j}}^2 - \sqrt{1 + 4n_{\mathrm{i,j}}^2\,\tilde{\sigma}_{\mathrm{i,j}}^4}}{2} \tag{4}$$

where $\zeta(\cdot)$ is softplus function, $\boldsymbol{E}_{\mathrm{e}}$ is the edge embedding generated in the first step, $\epsilon$ is a very small constant, $\mathrm{L_{mean}}(\cdot)$ and $\mathrm{L_{std}}(\cdot)$ are implemented by neural networks for estimating the mean and standard deviation respectively, $m_{\mathrm{i,j}} \in \boldsymbol{M}$ is the approximation of Binomial edge variable in the summary graph, and $\boldsymbol{M} \in \mathbb{R}^{B \times N_e}$ is the approximation of summary graph which strengthens the representation of real spatial or temporal relations.

*Step 3: Sleep relation intensity calculation.* The final step is to strengthen edge information and generate a relation intensity graph for downstream tasks. The relation intensities of brain temporal-spatial network during sleep is sparse based on existing research (Razi et al., 2017). Therefore,

generating a sparse graph based on edge embedding can not only highlight the representation of key relations, but also be more in line with the actual situation. We employ the Gaussian graph transformation approach which produces a sparse sleep relation intensity graph $\boldsymbol{G}$. The specific calculation is defined as follows:

$$\widetilde{\alpha}_{i,j} = \left( m_{i,j}^{std} \right) \times \varepsilon_{i,j} + m_{i,j} \tag{5}$$

$$s_{i,j} = \left( m_{i,j}^{mean} \right) \times \widetilde{\alpha}_{i,j} + \left( \widetilde{\alpha}_{i,j}^{std} \right) \times \left( \widetilde{\sigma}_{i,j}^{mean} \right) \times \varepsilon'_{i,j} \tag{6}$$

$$\bar{\alpha}_{i,j} = s_{i,j} \times \widetilde{\alpha}_{i,j} \tag{7}$$

$$\alpha_{i,j} = \zeta \left( L \left( \bar{\alpha}_{i,j} \right) \right) \tag{8}$$

where $m_{i,j} \in \boldsymbol{M}$ is the approximation of the edges of the summary graph obtained in Step 2, $\varepsilon$ and $\varepsilon'$ are the standard Gaussian random variable of the same dimension as $\boldsymbol{M}$, $\widetilde{\alpha} \in \mathbb{R}^{B \times N_e}$ is the Gaussian edge representation, $\boldsymbol{S} \in \mathbb{R}^{B \times N_e}$ is the task-related Gaussian variable, $\tilde{\sigma}_{i,j}$ is calculated in Eq.(3), $std$ is the standard deviation, $mean$ is the mean value, $\bar{\alpha}$ is the Gaussian transformation map, $\alpha \in \mathbb{R}^{B \times N_e}$ is the final sleep relation intensity graph, $\zeta(\cdot)$ is softplus function, and $L(\cdot)$ is linear function. Afterwards, we utilize an attention mechanism-based method to convert the node embeddings into feature embeddings based on the sleep relation intensity graph as the output of Bayesian relation inference. The specific calculation is as follows:

$$\boldsymbol{E}_{out} = \boldsymbol{f}_{GAL} \left( \boldsymbol{E}_n, \alpha \right) \tag{9}$$

where $\boldsymbol{f}_{GAL}(\cdot)$ is the graph attention layer, $\boldsymbol{E}_n$ is the node embeddings of the input object, $\alpha$ is the sleep relation intensity graph, and $\boldsymbol{E}_{out} \in \mathbb{R}^{B \times N_v \times V_{out}}$ is the output node embeddings.

### 4.1.2 LEARNING OF BAYESIAN RELATION INFERENCE

We adopt variational inference to jointly optimise Baysian relation inference component. Inspired by VRNN (Chung et al., 2015), we can use the evidence lower bound (ELBO) for joint learning and inference. The details of how variational inference fits into our model are shown in the Appendix 3. Specifically, we use two random variables need to be optimised to describe the same random process data. The resulting objective is to maximize the ELBO:

$$\sum_{i=1}^{M} \left\{ KL \left( q \left( \tilde{\mathbf{A}}, \mathbf{S} \mid \mathbf{X}_{0:i} \right) \| p \left( \tilde{\mathbf{A}}, \mathbf{S} \mid \mathbf{X}_{0:i} \right) \right) - \mathbb{E}_{\tilde{\mathbf{A}}, \mathbf{S}} \left[ \log P \left( \mathbf{Y}_i \mid \mathbf{X}_i, \tilde{\mathbf{A}}, \mathbf{S} \right) \right] \right\} \tag{10}$$

where $\mathbf{S}$ is the task-related Gaussian variable, $\tilde{\mathbf{A}}$ is the Gaussian graph embedding, $q \left( \tilde{\mathbf{A}}, \mathbf{S} \mid \mathbf{X}_{0:i} \right)$ is the prior distribution, and $p \left( \tilde{\mathbf{A}}, \mathbf{S} \mid \mathbf{X}_{0:i} \right)$ is the posterior distribution. Since every variable in $\mathbf{S}$ are affected by $\tilde{\mathbf{A}}$ in Eq.(6), the KL term can be further written as:

$$\sum_{(i,j) \in \tilde{E}} \left\{ KL \left( \mathcal{B}(n, \tilde{\lambda}_{i,j}) \| \mathcal{B} \left( n, \tilde{\lambda}_{i,j}^{(0)} \right) \right) \right.$$
$$\left. + \mathbb{E}_{\tilde{\alpha}_{i,j}} \left[ KL \left( \mathcal{N} \left( \tilde{\alpha}_{i,j} * \mu_{i,j}, \tilde{\alpha}_{i,j} * \sigma_{i,j}^2 \| \mathcal{N} \left( \tilde{\alpha}_{i,j} * \mu_{i,j}^{(0)}, \tilde{\alpha}_{i,j} * \sigma_{i,j}^{(0)^2} \right) \right) \right] \right\} \tag{11}$$

Obviously the second term can be calculated, while the first term is tough to calculate because $n \to \infty$. According to Theorem 2 provided in DGP (Huang et al., 2020), we can convert it into an easy-to-solve value to approximate the calculation.

### 4.1.3 BAYESIAN TRANSFORMER MODULE

Transformer shows convincing results in various sequence modeling tasks (Li et al., 2021; Luo et al., 2021; Zhou et al., 2021b). However, traditional transformer does not have the ability to reason the relation between each pair of object nodes. This results in a lack of interpretability of the attention graph generated by transformer. Besides, the accuracy of traditional transformer is not good enough under some medical scenarios. The Bayesian relation inference component we proposed can infer

spatial-temporal relations efficiently. Hence, we integrate the Bayesian relation inference component mentioned in Section 4.1.1 with the transformer in a unified framework to simultaneously reason and model the spatial-temporal features of sleep EEG data.

*Bayesian spatial transformer.* To better construct the spatial functional connectivity of the brain and capture spatial features, we design the Bayesian spatial transformer. It contains two components: a Bayesian spatial relation inference component and a position feed-forward network, of which the core component is a Bayesian relation inference component. Specifically, the input of the Bayesian spatial transformer $S$ is the embeddings of $n$ spatial nodes. First we add position encoding to the input to introduce position information:

$$\tilde{S} = S + \mathbf{P}^{\text{ep}} \tag{12}$$

where $S \in \mathbb{R}^{(B \times N_t) \times N_s \times V}$ is the input spatial node embedding, $\mathbf{P}^{\text{ep}} \in \mathbb{R}^{(B \times N_t) \times N_s \times V}$ is the position encoding matrix, and $\tilde{S} \in \mathbb{R}^{(B \times N_t) \times N_s \times V}$ is the position-encoded spatial node embeddings. For the position encoding matrix, we follow the groundbreaking work (Vaswani et al., 2017) and use the sine and cosine functions to calculate.

We design a multi-head spatial Bayesian relation inference component which can reson about spatial relations to improve the representation learning ability of the model. The details of Bayesian relation inference component have been described in Section 4.1.1. The node embeddings with positional encoding are encoded as embeddings with spatial features after passing through the multi-head spatial Bayesian relation inference component and the feed-forward neural network layer:

$$\tilde{S}' = f_{\text{FNN}}\left(f_{\text{BSRI}}\left(\tilde{S}\right)\right) \tag{13}$$

where $\tilde{S}$ is the input node embedding, $f_{\text{BSRI}}(\cdot)$ is the Bayesian spatial relation inference module which inferences the spatial relation, $f_{\text{FNN}}(\cdot)$ is the feed-forward neural network layer, and $\tilde{S}' \in \mathbb{R}^{B \times N_t \times V_s}$ is the spatial relation intensity graph based features.

*Bayesian temporal transformer.* Similar to the Bayesian spatial transformer, in order to better capture the sleep transition rules, we design a Bayesian temporal transformer module to reason and model temporal features. The calculations of the temporal relation intensity graph based features are as follows:

$$\widetilde{T} = \tilde{S}' + \mathbf{P}^{\text{ep}} \tag{14}$$

$$\tilde{T}' = f_{\text{FNN}}\left(f_{\text{BTRI}}\left(\tilde{T}\right)\right) \tag{15}$$

where $f_{\text{BTRI}}(\cdot)$ is the Bayesian temporal relation inference component, $f_{\text{FNN}}(\cdot)$ is the feed-forward neural network layer, and $\tilde{T}' \in \mathbb{R}^{B \times N_{st}}$ is the temporal relation intensity graph based features. The classification results are generated by $\tilde{T}'$ after passing through a linear classification layer $f_{\text{C}}$:

$$\widehat{Y} = f_{\text{C}}\left(\tilde{T}'\right) \tag{16}$$

where $\widehat{Y}$ is the classification result of BSTT.

## 5 EXPERIMENTS

To verify the effectiveness of the Bayesian spatial-temporal transformer, we evaluate it on the Institute of Systems and Robotics, University of Coimbra (ISRUC) and Montreal Archives of Sleep Studies-SS3 (MASS-SS3) dataset.

### 5.1 DATASET

ISRUC dataset contains the PSG recordings from 100 adult subjects. Each PSG recording contains 6 EEG channels, 6 EOG channels, 3 EMG channels, and 1 ECG channel. MASS-SS3 dataset contains the PSG recordings from 62 adult subjects. Each PSG recording contains 20 EEG channels, 2 EOG channels, 3 EMG channels, and 1 ECG channel. The recordings are divided into time slices according to a sleep epoch every 30s. Sleep spacialists divide these time slices into five distinct sleep stages (W, N1, N2, N3, and REM) according to the AASM standard. There are also motion artifacts at the beginning and end of each subject's recording which are marked as unknown. We follow the previous study and remove these recordings (Supratak et al., 2017).

## 5.2 EXPERIMENT SETTINGS

We evaluate our model using $k$-fold cross-subject validation to ensure that the experiment results are correct and reliable. We set $k = 5$ in order to test all recordings efficiently. For optimizer, the Adam and Adadelta optimizer are deployed in MASS-SS3 and ISRUC dataset.

We use multi-channel EEG data for sleep staging to better capture brain network structure. Specifically, on the ISRUC dataset, we use all 6 EEG channels for experiments. In the MASS-SS3 dataset we use 19 channels of EEG signals. To comprehensively evaluate the Bayesian spatial-temporal transformer model and all baseline methods, we use accuracy (ACC), F1 Score, and KAPPA to evaluate the models. Specific information of the evaluation indicators and baseline models are shown in Appendix 2.

## 5.3 EXPERIMENT ANALYSIS

Table 1 and 2 indicate that the proposed model achieves the best performance compared to other baseline methods on both datasets. Specifically, the MCNN and MMCNN utilize the CNN model to automatically extract sleep features, while RNN based methods such as DeepSleepNet and TinySleepNet focus on the temporal context in sleep data, and model the multi-level temporal characteristics of the sleep process for sleep staging. Further, GraphSleepNet and ST-Transformer simultaneously model the spatial-temporal relations during sleep and achieve satisfactory results. However, GraphSleepNet and ST-Transformer cannot adequately reason the spatial-temporal relations, which limits the classification performance to a certain extent. Our Bayesian ST-Transformer uses the multi-head Bayesian relation inference component to infer spatial-temporal relations to better model spatial and temporal relations. Therefore, the proposed model achieves best classification performance on different datasets.

Table 1: Comparison of Bayesian spatial-temporal transformer and baselines on ISRUC dataset

| Method | ACC(%) | F1 Scroe(%) | KAPPA(%) |
|---|---|---|---|
| MCNN | 78.23 | 76.37 | 71.79 |
| MMCNN | 76.83 | 78.93 | 74.17 |
| MLP+LSTM | 76.08 | 72.87 | 69.16 |
| DeepSleepNet | 75.71 | 73.16 | 68.82 |
| TinySleepNet | 76.92 | 75.15 | 70.26 |
| U-Time | 75.52 | 71.04 | 67.92 |
| GraphSleepNet | 80.18 | 78.19 | 74.54 |
| ST-Transformer | 80.35 | 78.05 | 74.71 |
| **BSTT (Our)** | **81.96**[*] | **80.30**[*] | **76.78**[*] |

[*]indicates the significant differences between our model and other models ($p < 0.05$).

Table 2: Comparison of Bayesian spatial-temporal transformer and baselines on MASS dataset

| Method | ACC(%) | F1 Score(%) | KAPPA(%) |
|---|---|---|---|
| MCNN | 86.31 | 80.79 | 81.26 |
| MMCNN | 80.15 | 71.03 | 69.42 |
| MLP+LSTM | 86.31 | 81.55 | 80.12 |
| DeepSleepNet | 85.92 | 79.81 | 79.09 |
| TinySleepNet | 83.30 | 77.40 | 79.95 |
| U-Time | 85.23 | 78.36 | 77.94 |
| GraphSleepNet | 88.36 | 83.25 | 83.30 |
| ST-Transformer | 88.64 | 84.53 | 83.16 |
| **BSTT (Our)** | **89.50**[*] | **85.00**[*] | **84.37**[*] |

[*]indicates the significant differences between our model and other models ($p < 0.05$).

## 5.4 ABLATION EXPERIMENTS

To verify the effectiveness of each component in the Bayesian spatial-temporal transformer, we conduct ablation experiments to determine the impact of each module on the model's performance. Specifically, we design three variants of the Bayesian spatial-temporal transformer, including:

- Bayesian Spatial Transformer (BST), which removes the Bayesian temporal transformer module to determine the impact of modeling temporal relations on model performance.
- Bayesian Temporal Transformer (BTT), which removes the Bayesian spatial transformer module to determine the impact of modeling spatial relations on model performance.
- Spatial-Temporal Transformer (STT), which removes the relational inference component to determine the impact of Bayesian relational inference on model performance.

Figure 2 demonstrates that the performance of the variant models degrades after removing certain component or module. Among them, the removal of the relational inference component has the greatest impact on performance, which shows the importance of introducing relational inference to the sleep staging task. In addition, only modeling the spatial relation or temporal relation of the data also lead to a decrease in performance. It can be seen that modeling the spatial and temporal relation is helpful for the sleep staging task.

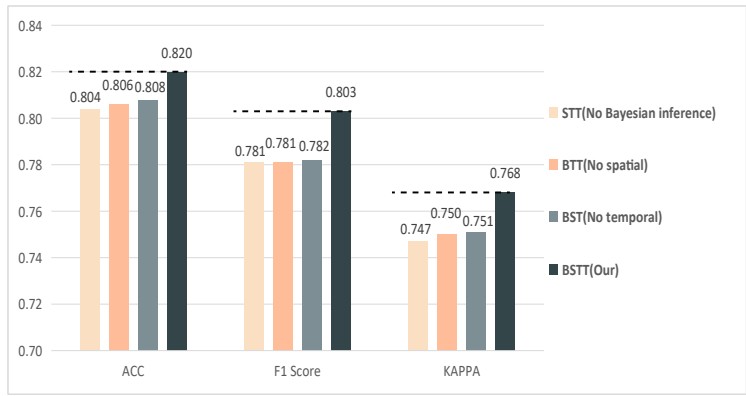

Figure 2: Ablation experiment results of Bayesian spatial-temporal transformer on ISRUC dataset.
.

## 5.5 VISUAL ANALYSIS

To verify the proposed Bayesian relational inference module can infer the spatial-temporal relations during sleep, we visualize and analysis the generated relational inference graphs.

### 5.5.1 VISUAL ANALYSIS OF SPATIAL RELATION INFERENCE

Some researches have shown that the functional connectivity of the brain varies during different sleep stages (Nguyen et al., 2018). In order to analyze the role of the Bayesian spatial inference component of our model, we visualize the spatial relation intensity graph between EEG signal channels at different sleep periods, as shown in Figure 3. The position of the nodes in the figure represents the position of the electrodes that output the EEG signals, and the edge is the relation intensity between the each two electrodes. We notice that during the NREM period, brain connectivity is significantly stronger in light sleep (N1, N2) than that during deep sleep (N3). It has been revealed that during light sleep, cerebral blood flow (CBF) and cerebral metabolic rate (CMR) are only about 3% to 10% lower than those of wakefulness while during deep sleep, these indexes have a significant decrease of 25% to 44% by previous study (Madsen & Vorstrup, 1991). Synaptic connection activity is directly correlated with CBF and CMR, which is consistent with our connection intensity graph. Madsen also reported that the level of brain synaptic activity during REM period is similar to that of the wake period, which matches our experimental findings.

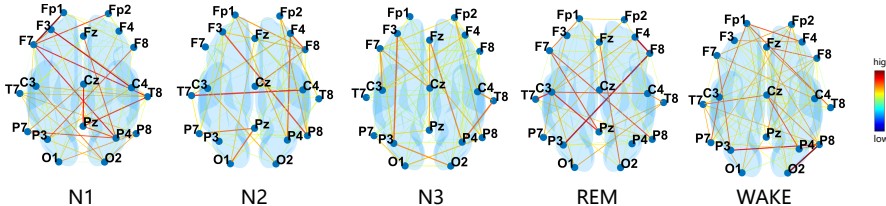

Figure 3: The graph shows the average of the brain spatial intensity over time. The spatial relation during the WAKE, REM and N1 period is strong, while the that during the N2 and N3 period is weak.

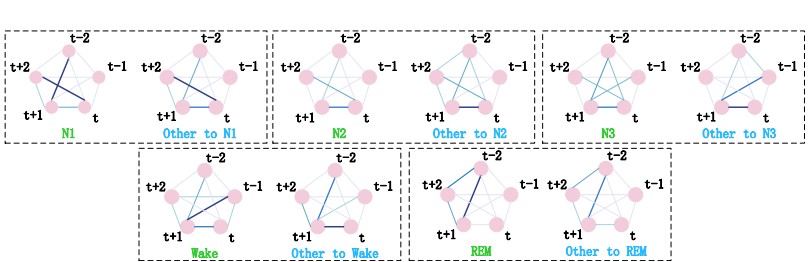

Figure 4: Intensity graphs of the temporal relation during different sleep periods and when sleep transitions occur.

### 5.5.2 VISUAL ANALYSIS OF TEMPORAL RELATION INFERENCE

To analyze the contribution of the Bayesian temporal inference component of our model for sleep staging, we visualize the time slice relation intensity graphs as shown in Figure 4. In each graph, the nodes represent time slices while edges represent the strength of the relation between time slices (Here, time slice represents the manually divided EEG signals of 30s duration.) The left graph of each graph pair (e.g. N1 marked green) represents the five time slices (nodes) are in the same sleep period. The one on the right (e.g. Other to N1 marked blue) represents the $t-2$ and $t-1$ time slices are in one sleep period while the other three are in another sleep period (Sleep period transition occurs between the $t-1$ and $t$ time slices.) In Figure 4, for each pair of graphs, the edges in the left graph are, on average, darker than those in the right one. This means that our model tends to think that EEG signals are more closely related to each other during a single sleep stage. Previous studies have shown that the stability of the unchanging period is stronger, and sleep instability is the basis of sleep transition (Bassi et al., 2009), which is consistent with our experimental results. Figure 4 also reports that when sleep transition occurs, the relation intensity between the $t$ and $t+1$ time slices are usually stronger. Similarly, temporal intensity between the $t-2$ and the last three time slices are usually weaker than that during the unchanging period, which is conducive to the interpretation of the sleeping transition.

## 6  CONCLUSION

We propose a novel Bayesian spatial-temporal transformer model for sleep staging. To our best knowledge, this is the first attempt to combine Bayesian relational inference with spatial-temporal transformer for sleep staging. Our BSTT constructs spatial-temporal relations through Bayesian relational inference and applies the transformer architecture to capture brain spatial-temporal features for sleep staging. The results show that BSTT can effectively improve the model performance and achieve state-of-the-art results. In addition, visual analysis presents that the relation intensity graphs generated by the Bayesian relation inference have certain interpretability, which is consistent with the existing research and helps to reveal the potential working mechanism of our model. Besides, the proposed BSTT is a general-framework which can inference the spatial-temporal relations of EEG data and perform satisfactory data forecasting. In the future, the proposed method can be used for other EEG tasks, such as emotion recognition or motor imagery classification.

## 7 REPRODUCIBILITY STATEMENT AND ETHICS STATEMENT

We provide an open-source implementation of our BSTT and other baseline models. The code of BSTT is available at: `https://github.com/YuchenLiu1225/BSTT/tree/main/BSTT`. Please check the Appendix 7 for links of the baseline methods.

The authors do not foresee any negative social impacts of this work. All authors disclosed no relevant relationships.

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
