# OpenReview forum: "BSTT: A Bayesian Spatial-Temporal Transformer for Sleep Staging"
_ICLR.cc/2023/Conference — ICLR 2023 poster_

### Official Review · Reviewer_USPW · 2022-10-24

**Confidence:** 4
**Correctness:** 3
**Technical Novelty And Significance:** 3
**Empirical Novelty And Significance:** 3
**Recommendation:** 8

**Clarity, Quality, Novelty And Reproducibility:**

Please see the detailed comments above.

*Clarity: Fair

*Quality: Good

*Novelty: Good

*Reproducibility: Good

**Strength And Weaknesses:**

*Strength

- The problem of modeling temporal and spatial relations in an interpretable way for sleep staging is well motivated and explained.

- This paper introduces the Bayesian relational inference in detail in terms of graphics, figures and formulas. How data and output flow are explained in the Bayesian relational inference step by step.

- This paper compares some recently proposed baselines, and obtains the results of SOTA on two datasets, and the accuracy is improved by two percentage points.

- The ablation experiments were valuable in demonstrating the impact of the proposed methods.


*Weaknesses

- The authors should add a pipline diagram so that readers can understand the overall process.

- In the problem definition of preliminaries, the input data only reflects temporal information and doesn't very reflect spatial information. The problem defined is more like a regular time series problem, which is difficult to directly contact with the spatial-temporal relations emphasized above.

- Some components used in the paper aren't introduced or cited in detail, such as the multi-head spatial Bayesian relation inference component.


**Summary Of The Paper:**

This paper proposes a novel model named BSTT for sleep staging. BSST integrates the transformer and Bayesian relation inference to simultaneously model spatial-temporal relations and effectively capture the spatial-temporal features of the brain. Bayesian relation inference, the core component of BBST, comes in two forms, Bayesian temporal relation inference and Bayesian spatial relation inference and is proposed to infer spatial-temporal relations of brain during sleep in the service of capturing spatial-temporal relations. Experimental results show that BSTT achieves the state-of-the-art in multiple sleep staging datasets.


**Summary Of The Review:**

It is interesting to infer the space-time relationship during sleep. This work provides some meaningful insights for sleep staging based on deep learning. Some details of the presentation need minor modification.

---

### Official Review · Reviewer_sN5h · 2022-10-25

**Confidence:** 3
**Correctness:** 3
**Technical Novelty And Significance:** 2
**Empirical Novelty And Significance:** 3
**Recommendation:** 5

**Clarity, Quality, Novelty And Reproducibility:**

Clarity:
The motivation is not clearly stated in the Bayesian transformer usage. Spatial and temporal modeling are combined, requiring more detailed explanations because there can be other approaches to combine spatial and temporal modeling, such as simple concatenation. Also, the connection of the theorem to vae should be clarified more thoroughly. Minor question: On page 8, what does “five-time slices are in the sleep period” mean?

Quality:
The organization of this paper can be improved, especially in the algorithm parts. Also, in Appendix, the proof seems strange. In page 15, $(1- \lambda) < (1- n \lambda)$ should be explained. More editing is required to improve the paper.

Novelty and reproducibility:
Somewhat novel. It seems that this paper is an application paper. Bu the applications seem successful. In the algorithm aspect, the novelty is degraded. Also, it seems that reproducibility can be possible. More repetition is required to validate the algorithms.


**Strength And Weaknesses:**

Pros:
The performance is better than other algorithms, and the experiment is conducted compared with various algorithms. Also, the usage of the Bayesian transformer is interesting. This paper is strong in the application aspects.

Cons:
In algorithm aspects, there are many issues. At first, the authors used the Bayesian transformer module. Many people know the transformer model, but some readers cannot be familiar with Bayesian transformers. It is better to add a more comprehensive explanation for the Bayesian transformer and the motivation for using this module. Next, there are some theoretical statements in the paper and Appendix. However, the organization of these materials is not well presented.


**Summary Of The Paper:**

In EEG signals to be used in sleep stage classification, there are multi-channels due to the location of brains (spatial), and previous sleep stages can affect the current sleep stage. This paper considers the Bayesian spatial-temporal transformer to classify the sleep stage. The authors proposed the Bayesian transformer module to be applied to capture the spatial-temporal characteristics of EEG. Experiments with the other algorithms were conducted, showing better performance. Also, visual analysis shows the relationship between channels or previous stages. The interpretation is reasonable in the area of the sleep study.

**Summary Of The Review:**

This paper is a good one in the application area. The contribution is somewhat low in the machine learning algorithm and learning representation. Also, I have concerns about more explanation, re-organization, and polishing the proofs.
This paper is a good one in the application area. In the machine learning algorithm and learning representation, the contribution is somewhat low. Also, I have concerns of more explanation, re-organization, and polishing the proofs.

---

### Official Review · Reviewer_s23K · 2022-10-25

**Confidence:** 3
**Correctness:** 4
**Technical Novelty And Significance:** 4
**Empirical Novelty And Significance:** 4
**Recommendation:** 8

**Clarity, Quality, Novelty And Reproducibility:**

Clarity: Section 4 needs improvement, but the experiments are presented well.
Quality and originality: The technical model is a good fit for the application problem, and performs well in experiments.

**Strength And Weaknesses:**

Strength: The experimental results are strong, and there are enough details to make the results reproducible with github code. The idea of spatial-temporal transformer (and some level of interpretability with the graphs) is very promising for this application domain.

Weakness: The presentation of this paper can be improved by a lot, especially in section 4.1.1. It's unclear what each variable means, and what their dimensions are. Also in Figure 1, should one of the two transformers be spatial?
In results, what were the standard deviation from cross validation? Finally, in Table 8, the results in the last row are bolded even when it's not the highest F1 score compared to the baselines.

**Summary Of The Paper:**

The paper proposes a combination of Bayesian relational inference with spatial-temporal transformer for automatic sleep scoring. In addition to testing the new deep learning model's accuracy in sleep scoring, it provides ablation experiment results and visualizations of relation intensity graphs that come from this model as a by product.

**Summary Of The Review:**

The idea of Bayesian spatial-temporal transformer is interesting and seems appropriate for sleep scoring. Also the experimental results are very strong. But the paper should be written more clearly.

---

### Official Review · Reviewer_dJC8 · 2022-10-25

**Confidence:** 4
**Correctness:** 3
**Technical Novelty And Significance:** 3
**Empirical Novelty And Significance:** 3
**Recommendation:** 5

**Clarity, Quality, Novelty And Reproducibility:**

The clarity aspect is missing as mentioned in previous section but the paper covers all the other aspects well and provides a good codebase for reproducibility.

**Strength And Weaknesses:**

Strengths:
- Surpasses state-of-the-art for sleep staging
- Shows some spacial interpretability


Weaknesses:
- Improvements are minor (<1% for MASS dataset)
- Figure 1 both transformers are identical.
- Section 4, the method is not well written. The terminology is not explained.
- Figure 4 is not explained well and it is unclear how it provides interpretation.

**Summary Of The Paper:**

The paper proposes a method for sleep staging based on Bayesian spatial-temporal transformer.

**Summary Of The Review:**

The paper has some novel contributions but does not outperform existing methods by any significant margin. Moreover, the writeup needs a lot of work. The terms are not explained. One of the two interpretations is either not very convincing or not explained well.

---

### Decision · Program_Chairs · 2023-01-20

**Decision:**

Accept: poster

**Justification For Why Not Higher Score:**

Some of the authors didn't give very high scores, though they seemed to accept the authors' answers.

**Justification For Why Not Lower Score:**

My own judgment on the merits of the paper.

**Metareview: Summary, Strengths And Weaknesses:**

This paper introduces a novel neural network architecture that combines Bayesian relational inference [Huang et al, ICML 2020] with spatial-temporal transformers. The model is well-motivated by the application domain, sleep staging, which is the automated classification of sleep stages from EEG signals. EEG signals exhibit strong spatial and temporal dependencies, which the model is well-designed to capture. The paper gives convincing experimental results both in terms of accuracy and interpretability. It is likely the model could be adapted for other spatiotemporal inference problems so should be of interest beyond the particular application domain.

**Note From Pc:**

if the above contains the word "oral" or "spotlight" please see: "oral" presentation means -> notable-top-5% and "spotlight" means -> notable-top-25%. As stated in our emails, we are disassociating presentation type from AC recommendations